# Independent *COL17A1* Variants in Cats with Junctional Epidermolysis Bullosa

**DOI:** 10.3390/genes14101835

**Published:** 2023-09-22

**Authors:** Sarah Kiener, Heather Troyer, Daniel Ruvolo, Paula Grest, Sara Soto, Anna Letko, Vidhya Jagannathan, Tosso Leeb, Elizabeth A. Mauldin, Ching Yang, Ana Rostaher

**Affiliations:** 1Institute of Genetics, Vetsuisse Faculty, University of Bern, 3001 Bern, Switzerland; sarah.kiener@unibe.ch (S.K.); anna.letko@unibe.ch (A.L.); vidhya.jagannathan@unibe.ch (V.J.); 2Dermfocus, University of Bern, 3001 Bern, Switzerland; sara.soto@unibe.ch; 3Oradell Animal Hospital, Paramus, NJ 07652, USA; htroyer@oradell.com (H.T.); druvolo@oradell.com (D.R.); 4Institute of Veterinary Pathology, Vetsuisse Faculty, University of Zurich, 8057 Zurich, Switzerland; grest@vetpath.uzh.ch; 5Institute of Animal Pathology, Vetsuisse Faculty, University of Bern, 3001 Bern, Switzerland; 6School of Veterinary Medicine, University of Pennsylvania, Philadelphia, PA 19104, USA; emauldin@vet.upenn.edu (E.A.M.); ching.yang@liu.edu (C.Y.); 7College of Veterinary Medicine, Long Island University, Brookville, NY 11548, USA; 8Clinic for Small Animal Internal Medicine, Vetsuisse Faculty, University of Zurich, 8057 Zurich, Switzerland; arostaher@vetclinics.uzh.ch

**Keywords:** *Felis catus*, WGS, dermatology, skin, JEB, splicing, precision medicine, animal model

## Abstract

Epidermolysis bullosa (EB), characterized by defective adhesion of the epidermis to the dermis, is a heterogeneous disease with many subtypes in human patients and domestic animals. We investigated two unrelated cats with recurring erosions and ulcers on ear pinnae, oral mucosa, and paw pads that were suggestive of EB. Histopathology confirmed the diagnosis of EB in both cats. Case 1 was severe and had to be euthanized at 5 months of age. Case 2 had a milder course and was alive at 11 years of age at the time of writing. Whole genome sequencing of both affected cats revealed independent homozygous variants in *COL17A1* encoding the collagen type XVII alpha 1 chain. Loss of function variants in *COL17A1* lead to junctional epidermolysis bullosa (JEB) in human patients. The identified splice site variant in case 1, c.3019+1del, was predicted to lead to a complete deficiency in collagen type XVII. Case 2 had a splice region variant, c.769+5G>A. Assessment of the functional impact of this variant on the transcript level demonstrated partial aberrant splicing with residual expression of wildtype transcript. Thus, the molecular analyses provided a plausible explanation of the difference in clinical severity between the two cases and allowed the refinement of the diagnosis in the affected cats to JEB. This study highlights the complexity of EB in animals and contributes to a better understanding of the genotype-phenotype correlation in *COL17A1*-related JEB.

## 1. Introduction

Epidermolysis bullosa (EB) represents a group of rare genetic disorders characterized by loss of dermoepidermal integrity (blistering) in haired skin and mucous membranes in response to friction or trauma [1,2]. The prevalence of inherited forms of EB in humans ranges from 19 to 67 per million people in different countries [3]. The prevalence in veterinary medicine is unknown due to its rarity and lack of large database registries. In human medicine, EB was first described by Koebner in 1886 [4], followed by veterinary medicine, in the 1970s in sheep and dogs [5,6], then in the 1980s in horses and cattle [7,8] and in 1990s in cats [9,10]. Pathogenic variants affecting 16 distinct genes result in four main classical EB types: EB simplex (EBS), characterized by basal or suprabasal clefting; junctional EB (JEB), characterized by clefting within the lamina lucida of the basal membrane zone (BMZ); dystrophic EB (DEB), characterized by separation within or below the lamina densa of the BMZ; and Kindler syndrome (KS). characterized by poikiloderma, trauma-induced skin blistering, mucosal inflammation, and photosensitivity [3]. EB types are further subclassified into more than 30 subtypes, according to their clinical severity (severe, intermediate, and localized, with/without extracutaneous involvement) and the underlying molecular defect [1,3]. Related to these (sub)types, several phenotypes have been described, with complications ranging from localized skin blistering, deranged wound healing, sepsis, blindness, dehydration, and gastrointestinal problems (e.g., malnutrition, esophageal stenosis) to death. In humans, the prognosis of EB varies depending on the subtype of the disorder. While some patients have normal life expectancies, others can be at major risk of death during the first few years of life [11]. Mortality related to EB may occur at different ages. It is common in early infancy in cases of severe JEB in humans [11], while in severe recessive DEB, it is more common in young adulthood [12]. Likewise, animals are euthanized or die during the first months of life. Reports of survival for months or years in animals with EB are rare [13].

Treatment options for patients with EB are limited. Allogeneic stem cell transplantation (SCT) has been proposed as a therapeutic approach, but the experience is still limited [11,14]. Therefore, the primary aim of the treatment is to protect the skin from unnecessary trauma and prevent complications, which includes nutritional support, pain management, infection control, and promotion of healing [11].

In contrast to humans, where more than 1000 variants in at least 16 genes have been associated with EB [1], molecular studies are rarely reported in domestic animals. EBS was associated with variants in *PLEC* in a litter of dogs [15], *KRT5* in cattle [16,17] and a dog [18], and *KRT14* in a cat [19]. DEB was associated with variants in *COL7A1* in cattle [20] and dogs [21]. JEB was associated with variants in all three genes encoding subunits of laminin 332, *LAMA3* in dogs [22], horses [23] and cattle [24], *LAMB3* in dogs [25], and *LAMC2* in horses [26], sheep [27], and cattle [28].

In human JEB patients, variants in the *COL17A1* gene encoding the three identical subunits of collagen type XVII often result in relatively mild phenotypes [29]. In a single dog with a severe JEB phenotype, collagen XVII deficiency was demonstrated utilizing immunofluorescence, but the underlying genetics was not investigated [30]. Collagen type XVII, also known as BP180/BPAG2, is a type-II-oriented transmembrane collagen composed of 3 identical 180 kDa α-chains [31]. It is one of the hemidesmosomal components of basal keratinocytes. Collagen type XVII links keratin intermediate filaments to the underlying dermis via plectin, BP230, laminin 332, and collagen type VII [32].

This study documents two spontaneous JEB cases in cats with independent variants in the *COL17A1* gene, leading to clinical phenotypes of different severity.

## 2. Materials and Methods

### 2.1. Ethics Statement

All cats in this study were privately owned. Skin biopsies and blood samples for diagnostic purposes were collected with the consent of their owners. The collection of blood samples from control animals for WGS analyses was approved by the Cantonal Committee for Animal Experiments (Canton of Bern, Switzerland; permit BE94/2022). A skin sample for transcript analysis from another control cat was collected post-mortem. All animal experiments were done in accordance with local laws and regulations.

### 2.2. Histopathological Examinations

Punch biopsies of the skin lesions from haired skin, paw pads, and oral mucosa were collected and submitted for histopathological examination. Biopsies were fixed in 10% neutral-buffered formalin, routinely processed, and stained with hematoxylin and eosin (H&E) or periodic acid-Schiff reagent (PAS).

### 2.3. Animals for Genetic Analyses

The study included two unrelated domestic shorthair cats diagnosed with epidermolysis bullosa (index cases). Blood samples of both cases and a skin sample of case 2 were available. Additionally, a skin sample from an EB-unaffected unrelated control cat was collected for transcript analysis. This was a 13-year-old domestic shorthair cat that was euthanized due to metastatic mammary carcinoma.

### 2.4. DNA Isolation and Whole Genome Sequencing

DNA was isolated from EDTA blood on a Maxwell RSC 48 instrument using the Maxwell RSC Whole Blood DNA Kit (Promega, Dübendorf, Switzerland). The genomes of both affected cats were sequenced at 32× (case 1) and 30× (case 2) coverage using PCR-free libraries with ~400 bp insert size on an Illumina NovaSeq 6000 instrument (Next Generation Sequencing Platform of the University of Bern). Paired-end 2 × 150 bp reads in fastq files were processed into binary alignment map (bam-file) with respect to the F.catus_Fca126_mat1.0 genome reference assembly (GCF_018350175.1), and single nucleotide variants and small indels were called using GATK HaplotypeCaller [33]. The SnpEff software v5.0e [34] was used together with NCBI annotation release 105 for functional effect prediction of the called variants. The accession numbers of the sequence data of both cases were deposited in the European Nucleotide Archive and are listed in Appendix A.

### 2.5. Variant Filtering

We individually filtered for private homozygous or heterozygous variants in both index cases by comparing their genomes to a control cohort comprising 82 publicly available WGS data from genetically diverse cats (Appendix A). The allele frequency threshold in the control cohort was set to 0. In a second step, protein-changing variants were filtered, considering variants with a SnpEff predicted impact of “high” or “moderate”, as well as “low” impact variants affecting the splice region [34]. In a last step, we filtered for protein-changing variants in 16 known EB candidate genes [35].

### 2.6. Targeted Genotyping

The candidate variants detected by Illumina WGS of case 1 and case 2 were validated using Sanger sequencing. PCR products were amplified from genomic DNA using AmpliTaqGold360Mastermix (Thermo Fisher Scientific, Waltham, MA, USA) with forward and reverse primers (Appendix A) and standard PCR protocols with 30 cycles. The purified amplicons were sequenced on an ABI 3730 DNA Analyzer (Thermo Fisher Scientific, Reinach, Switzerland) and analyzed with the Sequencher 5.1 software (Gene-Codes, Ann Arbor, MI, USA).

### 2.7. RNA Extraction and RT-PCR

RNA was extracted from skin samples of case 2 and an unaffected control cat using the RNeasy Fibrous Tissue Mini Kit (Qiagen, Hilden, Germany). Tissue processing, RNA extraction, and reverse transcription into cDNA were performed as previously described using the SuperScript IV Reverse Transcriptase kit (Thermo Fisher Scientific) [36]. A 518 bp PCR product was amplified with AmpliTaqGold360Mastermix (Thermo Fisher Scientific) from cDNA using a forward primer spanning exon boundaries 7 and 8, and a reverse primer spanning exon boundaries 12 and 13 of the *COL17A1* gene (Appendix A). PCR amplicons were then Sanger sequenced and analyzed as described above.

### 2.8. Bioinformatic Prediction of Functional Effects on Splicing

The SpliceAI web server was used for the prediction of the functional impact of genomic variants on splicing [37,38]. This tool assesses the impact of human variants and uses the human genome reference sequence and data on human transcripts. The sequence of the feline 5′-splice site after exon 45 of the *COL17A1* gene is perfectly conserved with the corresponding human sequence (positions −3 up to +10 of the splice site). A prediction for the splice site following exon 10 could not be performed as feline and human sequences are not highly conserved.

## 3. Results

### 3.1. Clinical and Histopathological Phenotype Characterization

Case 1: A 3-week-old, female, intact American Shorthair cat was presented to an animal emergency room after being found outside by a good Samaritan. The kitten was surrendered to a rescue organization and hospitalized for nutritional support, treatment of hypothermia and dehydration, and multifocal areas of integumentary crusting and hypotrichosis (Figure 1a). On intake, a skin fungal culture was performed due to suspicion of cutaneous dermatophytosis. The kitten was started on empirical amoxicillin (Amoxi-Drops, Zoetis, Parsippany-Troy Hills, NJ, USA; 11 mg/kg PO twice daily) and topical miconazole (MicaVed Lotion 1%, Vedco, St. Joseph, MO, USA; daily). Empricial antihelmintic treatment consisted of a five-day course of fenbendazole (Panacur, Merck, Rathway, NJ, USA; 50 mg/kg PO daily). After 1-week of hospitalization, the kitten was brighter, eating vigorously, and interactive, but the integumentary lesions did not resolve. On physical examination, the kitten had bilateral mucoid ocular discharge and ceruminous exudate in the ear canals. The kitten also had hair loss with adherent scale and crust around the eyes, the dorsal muzzle, and dorsal cranium. The nailbeds were erythematous. Ulcers were present on the tongue and hard palate. The epidermis of the dorsal head easily sloughed with gentle pressure during cleaning (positive Nikolsky sign). Topical silver sulfadiazine cream (Ascend Laboratories, LLC, Parsippany, NJ, USA) was applied to the resulting ulcers on the dorsal head to facilitate re-epithelialization and to prevent secondary infections. Amoxicillin was also discontinued, and amoxicillin trihydrate/clavulanate potassium (Clavamox, Zoetis; 25 mg/kg PO twice daily) and itraconazole (Itrafungol, Elanco, Greenfield, IN, USA; 11 mg/kg PO daily) were started in its place.

The haired skin ulcers healed; however the oral ulcers were static (Figure 1b,c). The kitten was weaned onto commercial kitten canned food (Purina^®^ Pro Plan^®^ Veterinary Diets Kitten Canned Formula, Nestle Purina, St. Louis, MO, USA), and began to show signs of dysphagia (dropping food, chewing on one side of the mouth). A second course of Clavamox and itraconazole were given and a confirmatory dermatophyte PCR was negative. The kitten remained hospitalized for approximately 4-weeks before being discharged to a foster home.

Onychomadesis was noted at approximately 9-weeks of age, when the kitten was seen by an emergency clinician for a swollen right front paw (Figure 1d). The kitten began to exhibit avoidance behavior and gait abnormalities associated with its foot lesions. On examination, no nails were observed on any of the paws and the right forepaw had swelling with blistering lesions in the nail bed. A second dermatophyte PCR was submitted and a third course of itraconazole and Clavamox were initiated with the addition of topical chlorhexidine, acetic acid, and ketoconazole wipes (Mal-A-Ket Wipes, Dechra, Northwich, UK), an application of selamectin (Revolution, Zoetis), and an Elizabethan collar to prevent self-traumatization. The kitten was treated with compounded buprenorphine (Covetrus^®^ Roadrunner Pharmacy, Phoenix, AZ, USA, 0.01 mg/kg oral transmucosal, three times daily) intermittently for analgesic support during nursing care.

The second dermatophyte culture and PCR were negative, and slow improvements in the skin and paw lesions were noted. A recheck examination performed 2-weeks later, however, revealed the oral, nail bed, and integumentary lesions to be static. Red-tinged vesicles were observed at each nail bed instead of mature nails. Persistent ulcerations were still present on the hard palate, at the lip commissures, on the tongue, and on the concave surface of both pinnae (Figure 1e,f). The kitten was admitted to the hospital again with a restricted environment to prevent self-harm and was offered only paper litter and soft food. A Calicivirus PCR using an oropharyngeal and conjunctival swab and SNAP FIV/FeLV Combo Test (IDEXX, Westbrook, ME, USA) were performed at this time and were negative. A complete blood count and biochemistry panel also revealed no significant findings.

In addition to collecting punch biopsy samples for histopathology, tissue samples were also collected for a second Calicivirus PCR and systemic mycoses PCR, which were both negative. Skin cytology of the lesions, prior to biopsy collection, were negative for microorganisms, but an aerobic culture and sensitivity panel grew a secondary methicillin-resistant, coagulase-negative Staphylococcus species, which was susceptible to and treated with clindamycin (ClindaCure, Vedco; 17 mg/kg PO twice daily).

Histopathologic examination of the haired skin from the pinna and limb revealed subepidermal clefts with mild dermal inflammation and crusts containing cocci (Figure 2). The mucosa of the tongue was detached and demonstrated regional coagulative necrosis.

Due to the poor prognosis, the low potential for adoption, and the continuous development of painful ulcers on the tongue and extremities, the patient was ultimately euthanized. Requests for a necropsy were denied by the rescue organization in charge of the patient’s care.

Case 2: A 4-month-old male, intact, European Shorthair outdoor cat was referred for onychomadesis and skin and oral erosions. The condition had not responded to cefalexin (Cefacat, Biokema, Crissier, Switzerland; 25 mg/kg twice daily for 2 weeks) and prednisolone (2 mg/kg daily for 10 days). The cat was fed on a complete commercial diet and was otherwise in good general condition. On physical examination, multifocal erosions and crusts, mostly on the lips, hard palate, and ear pinnae (Figure 3a–d) were noted. Several claws were either dystrophic (Figure 3e) or missing, and the associated nail beds were eroded and covered by dark brown crusty material (Figure 3f). Additionally, some paw pads showed focal erosions with re-epithelization (Figure 3g). The affected feet were painful upon manipulation. Cytological evaluation of impression smears acquired from the pinnae and feet did not reveal any micro-organisms. Wood’s lamp examination and a fungal culture from skin scrapings were negative for dermatophytes. The complete blood count and biochemical results were all within normal limits, and the cat tested negative for FIV, FeLV, and Calicivirus. Based on the clinical data, a diagnosis of hereditary EB was suspected.

Histopathological features included multifocal subepidermal clefts and vesicles (Figure 4a) with a minimal presence of erythrocytes and inflammatory cells. The periodic acid-Schiff (PAS) staining positive basement membrane was clearly visible only on the dermal side (Figure 4b). The basal cells in areas with clefts showed variable condensed nuclei and hypereosinophilic cytoplasm, and the dermis was unremarkable.

The patient was eventually adopted by one of the authors (AR) at the age of 1 year. At the time of writing, the cat was still alive at the age of 11 years. Its JEB phenotype was associated with a good long-term outcome, as the clinical signs diminished in severity over time. The oral ulcers, although still present, reduced approximately 20% in size and the patient showed onychomadesis only in single claws every 2–3 months. The perioral, pinnal, and paw pad lesions have been re-occurring on average 1–3 times per year within the observation period.

### 3.2. Genetic Analysis

We individually compared the genome sequence data of case 1 and case 2 to 82 genetically diverse control genomes and searched for plausible causative variants. Several filtering steps were performed. Heterozygous variants were considered for a potential dominant mode of inheritance and homozygous variants for a potential recessive mode of inheritance. We only retained case-specific private variants that were absent from all control cats. The last automated filtering step identified protein-changing private variants. Finally, a subjective inspection of the variants prioritized variants in 16 known EB candidate genes [35], based on the results of the clinical and pathological examinations (Table 1 and Appendix A).

The analyses identified a single candidate variant in case 1, but two possible variants in case 2. Case 2 had a heterozygous variant in *LAMA3*, which is a known candidate gene for EB. However, *LAMA3*-related forms of EB are recessively inherited, which would be incompatible with a mono-allelic *LAMA3* variant. Furthermore, three different protein impact prediction software tools classified the *LAMA3* variant as neutral or unknown (SNPs&GO, PON-2P, PredictSNP [39,40,41]. This prompted us to exclude the *LAMA3* variant as potential candidate for the EB phenotype in case 2.

The final result of the WGS analyses yielded very clear top candidate causative variants for both cases. Both identified variants were present in a homozygous state and located in the *COL17A1* gene (Figure 5). The variant in case 1 affected a canonical splice site; the variant in case 2 affected a splice region (Table 2).

The *COL17A1*:c.3019+1del variant in case 1 was predicted to disrupt the 5′-splice site of intron 45 with a probability of 99% [37]. As the cat was already euthanized at the time of the genetic investigation, no samples were available to investigate the predicted splice defect at the mRNA level.

The *COL17A1*:c.769+5G>A variant in case 2 affected the 5′-splice site of intron 10. We assessed the effects on splicing at the transcript level using RNA isolated from a skin biopsy. RT-PCR revealed that two different transcripts were expressed in the skin of case 2: the wildtype transcript and a mutant transcript with 7 nucleotides from intron 10 added to the normal exon 10, XM_006938156.5:r.769_770insguacaug (Figure 6). The 7 nucleotide insertion leads to a frameshift and introduction of a premature stop codon, truncating 1158 codons or 77.5% of the wild type *COL17A1* open reading frame.

## 4. Discussion

In this study, we investigated two unrelated cats with clinical and histopathological signs of EB. The clinical lesions in each cat were remarkably similar (cutaneous and oral ulcers, epidermal sloughing with minor trauma, and sloughing of the nails). Histopathology revealed an impaired ability of the epidermis to adhere to the dermis, which resulted in separation at the dermoepidermal junction. These features were not sufficient to subtype the EB; however, the PAS stain revealed the basement membrane on the dermal side of the cleft. This narrowed the potential subtypes to EBS and JEB. Case 1 was euthanized due to the severity of the oral lesions; the lesions in case 2 were not as severe. Cat 2 is alive at the time of the writing and under the direct care of a veterinarian.

EB is a rare disease also in the feline species, with few cases reported at the clinico-pathological level. Previously reported cases were characterized by onychomadesis and localized or generalized skin and oral ulcerations, resulting in either mild or severe phenotypes, as seen in our cases [9,13,19,42]. A *KRT14* variant was identified in a single cat with EBS [19]. In another study involving two cats diagnosed with JEB, *LAMC2* and *LAMB3* were identified as possible candidate genes by indirect immunofluorescence, but the underlying genetics was not reported [13].

Whole-genome sequence analyses of the two affected cats described herein identified two independent variants in *COL17A1*, a well-known functional candidate gene for JEB. The variant in case 1 was a homozygous single base-pair deletion at a canonical GT 5′-splice site, c.3019+1del. Unfortunately, no samples were available to assess the effect of the deletion at the transcript level. However, variants affecting the +1G at the 5′-splice site of other genes were previously described as pathogenic or likely pathogenic, most likely leading to a complete loss of gene function in dogs and humans with other inherited diseases [43,44,45]. These results corroborate an important function for the G at the +1 position of a canonical 5′-splice site. We hypothesize that the c.3016+1G>A variant in case 1 resulted in a complete loss of *COL17A1* function.

The second case had a homozygous single base pair exchange 5 nucleotides after the exon 10/intron 11 boundary, c.769+5G>A. A corresponding splice region variant at the +5 position has recently been described in the *CYB5R3* gene in a cat with methemoglobinemia [46]. The G at position +5 of the 5′-splice site occurs in 78% of human 5′-splice sites and normally forms a base-pair with a cytosine from the U1 snRNA [47]. Apart from the canonical GU dinucleotide, the +5 position exhibits the strongest sequence conservation in the intronic 5′-splice site, suggesting its important role in mRNA splicing [48].

The identified variant in case 2 resulted in an aberrant mRNA transcript; however, a wildtype transcript was also expressed. This provides a plausible explanation for the difference in phenotype severity between the two cases. We hypothesize that the presence of some functional protein in case 2 preserves residual collagen type XVII function, whereas in case 1, we suspect that no functional protein was expressed, resulting in the observed severe clinical phenotype.

The identification of two independent *COL17A1* variants in two unrelated cats with an EB phenotype provides strong support for the claimed causality of these two variants. However, our short read sequencing approach did not have 100% sensitivity to detect all variants. Large structural variants were not investigated in this study. Our analysis was limited to the genomic DNA level in case 1. In case 2, we identified a genetic variant at the genomic DNA level and confirmed its functional consequences at the transcript level. We did not experimentally confirm the predicted complete or partial protein deficiency in collagen type XVII in both cats.

To the best of our knowledge, we report the first two pathogenic *COL17A1* variants and their genotype-phenotype correlation in domestic animals. One earlier report characterized an autosomal recessive JEB in German Shorthaired Pointers with documented absence of collagen type XVII protein expression [30]. However, the molecular genetics in these dogs was not investigated.

## 5. Conclusions

We characterized two cats with JEB and identified two independent homozygous causal variants in the *COL17A1* gene. The splice site variant c.3019+1del in case 1 was predicted to lead to a complete deficiency in collagen type XVII and was associated with a severe clinical phenotype. The splice region variant c.769+5G>A in case 2 resulted in partial aberrant splicing with residual expression of wildtype transcript, which likely is responsible for the milder clinical phenotype in this cat.

## Figures and Tables

**Figure 1 genes-14-01835-f001:**
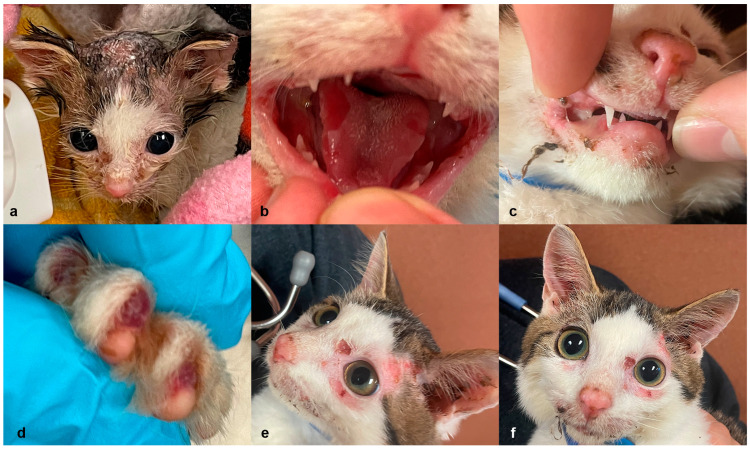
Details of the clinical phenotype of case 1. (**a**) Case 1 on day of initial presentation. (**b**) Erosions, ulcers, and crusts on the tongue and (**c**) lips. (**d**) Blistering vesicles with absence of nails on the paws. (**e**,**f**) Erosions, ulcers, and crusts on the periocular region and ear pinnae.

**Figure 2 genes-14-01835-f002:**
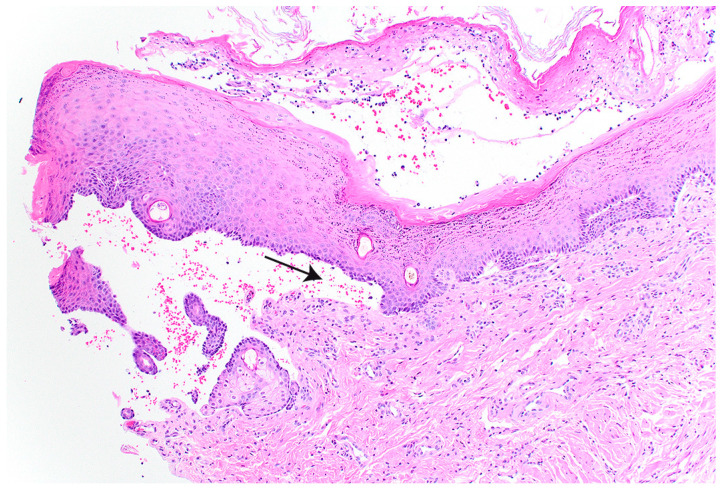
Histopathological findings in a skin biopsy from case 1. Subepidermal clefting is visible (arrow). H&E staining.

**Figure 3 genes-14-01835-f003:**
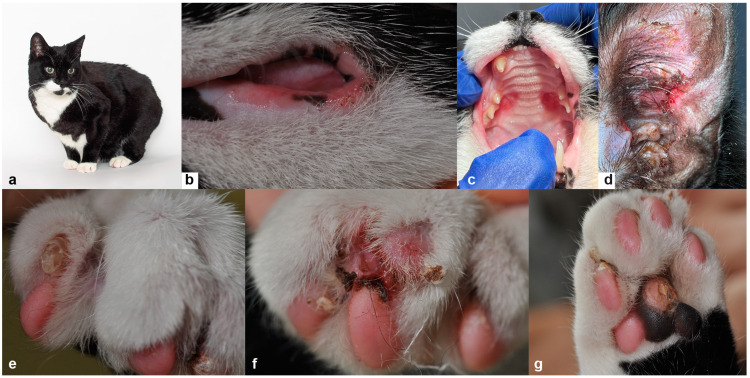
Details of the clinical phenotype of case 2. (**a**) Case 2 at 11 years of age. (**b**) Erosions and crusts on the lips, (**c**) hard palate and (**d**) ear pinnae. (**e**–**g**) Claws were dystrophic, and associated nail beds were eroded and covered with dark brown crusts.

**Figure 4 genes-14-01835-f004:**
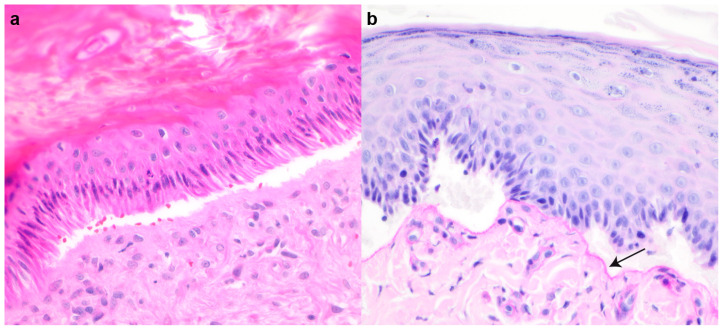
Histopathological findings in case 2. (**a**) Subepidermal cleft formation with low number of erythrocytes within the cleft. H&E staining. (**b**) Basal membrane (arrow) visible at the dermal side of the cleft. PAS staining.

**Figure 5 genes-14-01835-f005:**
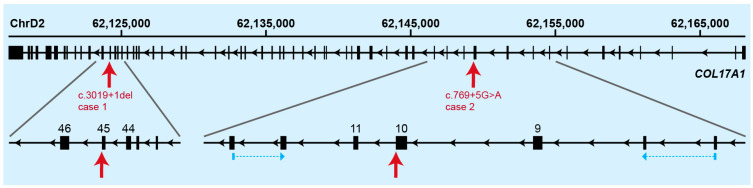
Overview of the genomic organization of the *COL17A1* gene. Exon numbering refers to the transcript with accession number XM_006938156.5. The positions of the variants of both investigated cats are indicated by red arrows. Enlarged details of the gene around both variants with the relevant exons are shown. Locations of the RT-PCR primers for cDNA analysis of the c.769+5G>A variant are indicated by blue arrows.

**Figure 6 genes-14-01835-f006:**
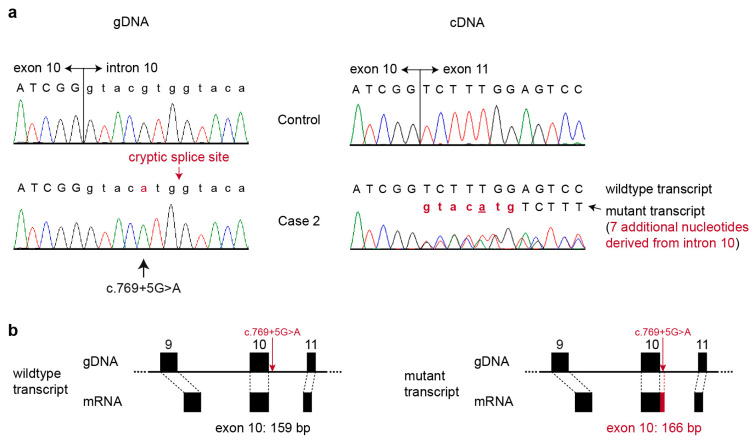
Effect of the *COL17A1*:c.769+5G>A variant in case 2 on splicing (**a**) Sanger electropherograms of the c.769+5G>A variant of case 2 and a control cat. The homozygous single nucleotide exchange at the genomic DNA level is indicated by a black arrow. At the cDNA level, the variant leads to partial aberrant splicing, resulting in the simultaneous expression of wildtype transcript and a mutant transcript with an additional 7 nucleotides derived from the beginning of intron 10. (**b**) Schematic representation of exons 9, 10, and 11 of the *COL17A1* gene and the wildtype and mutant transcripts. The c.769+5G>A variant is indicated at the genomic DNA level by a red arrow. The resulting wildtype and mutant transcripts are displayed.

**Table 1 genes-14-01835-t001:** Variants detected by whole genome sequencing of the two affected cats.

Filtering Step	Variants Case 1		Variants Case 2	
	Het	Hom	Het	Hom
All variants	6,364,334	5,790,564	6,976,390	5,784,208
Private variants (allele frequency of 0 in control cohort)	61,931	13,136	88,531	16,633
Protein-changing private variants	369	83	573	90
Protein-changing variants in 16 candidate genes	0	1	1	1

**Table 2 genes-14-01835-t002:** Variant designations of the identified *COL17A1* variants according to Human Genome Variation Society (HGVS) nomenclature.

Cat	HGVS-g ^1^	HGVS-c ^2^	HGVS-r ^3^	HGVS-p ^4^
	NC_058378.1	XM_006938156.5	XM_006938156.5	XP_006938218.3
Case 1	ChrD2:62,124,169del	c.3019+1del	r.spl?	n.a.
Case 2	ChrD2:62,149,308C>T	c.769+5G>A	r.[=,r.769_770insguacaug]	p.([=,p.Val257Glyfs*82])

^1^ g = linear genomic reference sequence; ^2^ c = coding DNA reference sequence; ^3^ r = RNA reference sequence (transcript); ^4^ p = protein reference sequence. * is the HGVS approved abbreviation for a termination codon.

## Data Availability

The accession numbers of the sequence data that are reported in this study are listed in Appendix A.

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
