# Peer review of "Independent COL17A1 Variants in Cats with Junctional Epidermolysis Bullosa"

_genes, 2023, doi:10.3390/genes14101835_

Round 1

Reviewer 1 Report

The authors reported two cases of EB in cats.  WGS was performed to detect pathogenic variants. The gene involved is COL17A1.

The study is well conducted and I have just some suggestions.

1) In "Materials and Methods" (lines 121-127), the authors performed genotyping validation with Sanger sequencing. In the figures, Sanger electropherogram was shown for the case 2 only (Figure 6).

2) In the case 2, although the experimental demonstration of the mild effect of the variant on the splicing is clear, it would be interesting to show the result of a prediction, whatever it is, performed with a bioinformatic tool. The bioinformatic tool can be also used to confirm the role of the case 1 variant (the mutation affects a canonical splice site).

Author Response

The authors reported two cases of EB in cats.  WGS was performed to detect pathogenic variants. The gene involved is COL17A1.

The study is well conducted and I have just some suggestions.

(1)

In "Materials and Methods" (lines 121-127), the authors performed genotyping validation with Sanger sequencing. In the figures, Sanger electropherogram was shown for the case 2 only (Figure 6).

Response: We confirmed both variants by Sanger sequencing. The Sanger electropherogram of case 1 unequivocally confirmed the single base deletion identified in the illumina WGS data. However, unfortunately, the Sanger sequences of case 1 had substantial background signals and are of insufficient quality for a figure in a publication.

We ask to not include a figure of the Sanger sequencing validation in the manuscript as this is a standard experiment. The main reason for including the Sanger sequencing of case 2 was the comparison of genomic DNA vs cDNA sequences. This is not applicable to case 1.

We ask for an editorial decision, whether the experiment must be repeated. We are happy to repeat and optimize the experiment, if requested by the editor. However, for this we will need ~3 weeks additional time.

(2)

In the case 2, although the experimental demonstration of the mild effect of the variant on the splicing is clear, it would be interesting to show the result of a prediction, whatever it is, performed with a bioinformatic tool. The bioinformatic tool can be also used to confirm the role of the case 1 variant (the mutation affects a canonical splice site).

Response: Unfortunately, we are not aware of a reliable bioinformatic tool that would allow to predict the effect of cat genomic variants on splicing. There are a number of such tools that are optimized and trained on human sequence (e.g. SpliceAI, SpliceVault). However, unfortunately the sequence at the end of the feline exon 10 and beginning of intron 10 harboring the c.769+5G>A variant is not well conserved with the human sequence. The “humanized” predictors can therefore not be applied to this variant.

The SpliceAI prediction for the c.3019+1del variant predicts the loss of the splice donor with a score of 0.99. The score can be interpreted as a probability of 99% that this variant changes the splicing. We added this information to the manuscript.

Reviewer 2 Report

This is a nice paper which finds two novel mutations in the COL17A1 gene in two cats with EB. I have some comments below which require addressing prior to publication.

Methods

- what breeds were the three cats used in the genetic analyses (lines 101-104)? I see this is in the table S1 but would be helpful to state here.

- what was the age of the control cat? Was the cause of death known?

- line 113. I find this confusing as in table S1 there is no mention of the control cat which is described in section 2.3? Or was this cat just used for a skin biopsy? Please could this be clarified along with the description of control cats on lines 91-92 (it is not clear if these are the 82 other cats you have WGS for?)

- line 112. Please provide more information on the variant calling pipeline so it is stand alone within this manuscript (and not relying on reference 33).

- section 2.4 what was the read length for the sequencing eg. 150 bp paired end? Was the sequencing performed in house or by a commercial supplier?

Results:

Lines 244-248. You describe filtering based on mode of inheritance and MAF as well as variant impact but in the methods only variant impact is described. What mode of inheritance did you use? What MAF cut off in the control population did you use?

Table 1 – by private variants do you mean they only occurred in that case and none of the controls? i.e. they had a MAF of 0 in the controls? What do K746 and K745 stand for?

Table 2 – please define HGVS-g, c, r, p

You say both a mutant and wildtype form were expressed but this is only shown via sequencing data. Is it not possible to show expression data for the alternative transcripts via PCR analysis?

Discussion:

The current discussion is very brief and should be expanded.

You state that the c.3019+1del has been described to be pathogenic and leads to a loss of gene function across multiple tissues and yet this cat didn’t appear to have defects in its skeleton, blood and muscle? Please expand this section.

You only looked for SNPs/small indels. Is it possible that there were other structural variants? Were the levels of COL17A1 protein affected in the skin samples? Please discuss other limitations of your study/conclusions.

Author Response

(1)

This is a nice paper which finds two novel mutations in the COL17A1 gene in two cats with EB. I have some comments below which require addressing prior to publication.

Response: Thank you for the positive evaluation of our manuscript.

(2)

Methods

- What breeds were the three cats used in the genetic analyses (lines 101-104)? I see this is in the table S1 but would be helpful to state here.

Response: We added the breed information to this paragraph (domestic shorthair, random-bred).

(3)

What was the age of the control cat? Was the cause of death known?

Response: It was a 13-year-old cat that was euthanized due to mammary carcinoma. We added the information in lines 106-107.

(4)

line 113. I find this confusing as in table S1 there is no mention of the control cat which is described in section 2.3? Or was this cat just used for a skin biopsy? Please could this be clarified along with the description of control cats on lines 91-92 (it is not clear if these are the 82 other cats you have WGS for?)

Response: We have slightly adapted the relevant paragraphs (2.1 and 2.3) to clarify which cats were used as controls for which experiments. The donor cat of the skin sample was only used as a control for transcript analysis.

(5)

line 112. Please provide more information on the variant calling pipeline so it is stand alone within this manuscript (and not relying on reference 33).

Response: We slightly revised the paragraph and give more details regarding the variant calling pipeline. We deleted the reference to our earlier publication.

(6)

section 2.4 what was the read length for the sequencing e.g. 150 bp paired end? Was the sequencing performed in house or by a commercial supplier?

Response: The read length was indeed 2x150 bp and the experiments were performed at the sequencing core facility of the University of Bern. We added this information to section 2.4.

(7)

Results:

Lines 244-248. You describe filtering based on mode of inheritance and MAF as well as variant impact but in the methods only variant impact is described. What mode of inheritance did you use? What MAF cut off in the control population did you use?

Response: We revised and expanded section 2.5 with the details on variant filtering. In brief, we considered dominant and recessive modes of inheritance. We then applied a hard filtering approach that required all controls to be clear of a potential pathogenic allele (private variants; MAF in controls = 0). We also revised the beginning of the results section (3.2) to make the filtering strategy more clear.

(8)

Table 1 – by private variants do you mean they only occurred in that case and none of the controls? i.e. they had a MAF of 0 in the controls? What do K746 and K745 stand for?

Response: Yes, private variants occur only in the case animals and are absent in the controls. This equals to MAF = 0 in the controls. We deleted “K746” and “K745” from Table 1. These were our internal laboratory identifiers.

(9)

Table 2 – please define HGVS-g, c, r, p

Response: We added footnotes explaining these abbreviations to the table.

(10)

You say both a mutant and wildtype form were expressed but this is only shown via sequencing data. Is it not possible to show expression data for the alternative transcripts via PCR analysis?

Response: We thank the reviewer for this valuable comment. The difference between mutant and wildtype transcript is only 7 bp. Unfortunately, our fragment size analysis (performed on an Agilent 5200 Fragment Analyzer that is based on capillary electrophoresis) was not able to clearly separate the two bands (518 bp vs 525 bp). We are therefore unable to provide a meaningful figure showing two bands corresponding to the two different transcripts. We show the Sanger sequencing electropherograms as the next best alternative instead.

(11)

Discussion:

The current discussion is very brief and should be expanded.

You state that the c.3019+1del has been described to be pathogenic and leads to a loss of gene function across multiple tissues and yet this cat didn’t appear to have defects in its skeleton, blood and muscle? Please expand this section.

Response: We revised and expanded the discussion. Our initial wording mentioning other tissues was probably confusing. We cite three publications describing splice variants at the +1 position in other diseases. These diseases affect other tissues but are also due to variants in other genes. We hope that the revised discussion is now clearer.

The protein expression of collagen type XVII is largely restricted to skin. A genetic defect in the COL17A1 gene in the affected cats is present in all cells. However, due to the tissue-specific expression and function of collagen type XVII, the clinical signs are restricted to the skin.

(12)

You only looked for SNPs/small indels. Is it possible that there were other structural variants? Were the levels of COL17A1 protein affected in the skin samples? Please discuss other limitations of your study/conclusions.

Response: We added a paragraph with limitations of the study to the discussion.